# Towards Internationalization: Exploring Economic Diplomacy in the Middle East (GCC)

Daniel Pontes [1], Vasco Santos [2,3,4,*], Orlando Samões [1], Shuangao Wang [5] and Ronnie Figueiredo [6,7,8,*]

1    Instituto de Estudos Políticos, Universidade Católica Portuguesa, Palma de Cima, 1649-023 Lisbon, Portugal; danielpontes326@gmail.com (D.P.); samoes@ucp.pt (O.S.)
2    Departamento de Economia, Gestão, Engenharia Industrial e Turismo (DEGEIT), Universidade de Aveiro, 3810-193 Aveiro, Portugal
3    CiTUR—Centre for Tourism Research, Development and Innovation, Polytechnic of Leiria, 2520-641 Peniche, Portugal
4    ISLA Santarém—Instituto Superior de Gestão e Administração, 2000-241 Santarém, Portugal
5    Beijing Academy of Science and Technology, No. 27, Beike Building, Haidian District, Beijing 100089, China; h130031@e.ntu.edu.sg
6    Centre of Applied Research in Management and Economics (CARME), School of Technology and Management (ESTG), Polytechnic of Leiria, 2411-901 Leiria, Portugal
7    Research Center in Business Sciences, NECE (UBI), 6200-209 Covilhã, Portugal
8    Spinner Innovation Centre (SIC), 2840-626 Lisbon, Portugal
*    Correspondence: vasco-rs@hotmail.com (V.S.); ronnie.figueiredo@ipleira.pt (R.F.)

**Abstract:** Internationalization is a crucial process for companies seeking growth and expansion in foreign markets, especially in the Middle East, where economies have been developing and diversifying business opportunities, seeing it as an attractive destination to expand their operations. This study presents a comprehensive analysis of the internationalization process of economic diplomacy by exploring the experiences of renowned Portuguese companies within the Gulf Cooperation Council. The Gulf Cooperation Council countries are integral players in the Middle East market, characterized as rentier states that are heavily reliant on oil and gas revenues and possess varying levels of economic and military strength, with Saudi Arabia and the UAE being the most prominent. The focus group method was applied in the qualitative research. It contributes to reinforcing the literature on internationalization processes, economic diplomacy, and the Middle East market. The findings provide valuable guidance to Portuguese companies, policymakers, and economic diplomats involved in promoting and facilitating international trade and investment.

**Keywords:** internationalization; economic diplomacy; Middle East; Gulf Cooperation Council; focus group

## 1. Introduction

The Persian Gulf represents an arena of intricate and conflicting interests, where Iran and Saudi Arabia emerge as the predominant regional forces. Meanwhile, the United States, with military dominance and a leaning toward Saudi Arabia rather than Iran, exerts significant influence (Garlick and Havlová 2020).

Current political and economic events result from the latest historical narratives of global significance, where international relations hold sway (Peternel and Grešš 2021). In addition, with the conclusion of the Cold War, economic diplomacy has garnered escalating attention within the realm of international relations research (Choi 2023).

Nowadays, the potential of the Middle East shows that the growth in industrialization of the Middle East countries, particularly the United Arab Emirates (UAE) and Qatar, has made this region one of the most attractive places for foreign companies to do business with countries that are part of the Gulf Cooperation Council (Karavar 2021).

For example, since the first discoveries of oil in the 1970s, these countries have shown economic development rates above the world average, going from tents to skyscrapers and competing with the world's most renowned business centers (Tobing and Virgianita 2020). In the present global political landscape, nations need to develop public diplomacy strategies to advance their national interests while navigating the challenges posed by ongoing global transformations (Triwahyuni 2022).

Amidst the shifts observed in the international order over the past decade, a set of practices has emerged that strategically harness financial and commercial dynamics to attain objectives beyond the economic realm. This signifies an escalating trend toward the politicization and utilization of international economic interactions for non-economic aims (Zelicovich 2023).

In addition, the economic element plays a substantial role in a nation's progress and its positioning on the global stage and has evolved into a pivotal gauge of a nation's influence (Khmel and Tykhonenko 2020).

The GCC countries exhibit a complex socio-political landscape that significantly influences the internationalization endeavors of companies operating within the region. While these nations boast substantial oil and gas reserves, rendering them rentier states with strong economic prowess, several socio-political dynamics and challenges shape the environment for international businesses: (a) Most GCC nations are governed by authoritarian regimes, where power is concentrated within ruling families or elites. Political stability often comes at the expense of limited civil liberties and political freedoms. This centralized governance structure can lead to bureaucratic hurdles and regulatory uncertainties for international companies seeking to establish operations. (b) Despite economic prosperity, many GCC countries grapple with high youth unemployment rates due to a rapidly growing young population. The challenge of integrating youth into the workforce is exacerbated by a reliance on expatriate labor and educational systems that may not align with evolving market demands. This demographic imbalance can affect social stability and consumer behavior, influencing market strategies for international companies. (c) The GCC region is characterized by diverse socio-cultural norms and practices deeply rooted in Islamic traditions. While modernization efforts have led to significant social changes, cultural sensitivities remain paramount. International businesses must navigate these cultural nuances carefully, particularly regarding marketing strategies, product localization, and workforce management. (d) Despite efforts to diversify their economies, GCC countries remain heavily dependent on oil and gas revenues. Fluctuations in global energy markets can have profound effects on government spending, investment strategies, and consumer purchasing power. Companies operating in sectors beyond energy must be resilient to economic volatility and adapt to shifting market conditions. (e) The GCC region is situated in a geopolitically sensitive area, marked by regional rivalries, conflicts, and security concerns. Tensions with neighboring countries, such as Iran and Yemen, as well as broader geopolitical dynamics involving major powers, can impact business operations and investment decisions. International companies must assess and mitigate geopolitical risks to safeguard their interests and ensure operational continuity. (f) GCC governments have increasingly implemented policies aimed at promoting nationalization and reducing reliance on expatriate labor. Companies operating in the region face evolving regulations related to employment quotas, visa requirements, and workforce localization. Compliance with these regulations while maintaining operational efficiency poses a significant challenge for international businesses. (g) Rapid urbanization and infrastructure development characterize many GCC countries, driven by ambitious economic diversification plans and population growth. While these initiatives present opportunities for investment and market expansion, they also pose challenges related to infrastructure capacity, urban planning, and environmental sustainability. International companies must align their strategies with evolving infrastructure priorities and contribute to sustainable development goals. The socio-political dynamics and challenges within the GCC region profoundly influence the internationalization process of companies. Understanding and navigating these complexi-

ties are essential for businesses seeking to establish a foothold in this dynamic and rapidly evolving market. Adaptability, cultural sensitivity, and strategic risk management are key factors for success in the GCC countries' business landscape (Rede Diplomática 2023).

This study presents a comprehensive analysis of the internationalization process of economic diplomacy by exploring the experiences of renowned Portuguese companies within the Gulf Cooperation Council. The research questions are as follows: *What variables in the Middle East make economic diplomacy effective, and in what ways has economic diplomacy contributed to the internationalization of companies?* The formulation of the research questions was prepared with the aim of finding a set of questions that would inquire into the perception of Portuguese companies looking for foreign markets, in this specific case, the Middle East, about the support of economic diplomacy in the context of their internationalization.

Since there is a lack of quantitative research in the field of internationalization and economic diplomacy, this article fills the gap with research based on the internationalization process of economic diplomacy, presenting the first results of a case study of companies in the Middle East. Therefore, the major contribution of this study lies in the intersection of the two constructs, namely economic diplomacy and internationalization processes, since research in this area and to this extent has never been advanced and carried out before.

The structure of the article follows that of a standard scientific paper. The introduction proposes the main aim of the research and identifies the gap in Middle East economic diplomacy. The literature review introduces the main works regarding the economic diplomacy context of companies' internationalization. The focus group method applied in the qualitative research contributes to reinforcing the literature on internationalization processes, economic diplomacy, and the Middle East market. The results present the case study analysis. The discussion and conclusion reveal the limitations and further prospects of the research in economic diplomacy, including the findings.

## 2. Background

Economic diplomacy is today the key to countries' external relations, an instrument of a state's foreign policy, and is not a new concept. However, in the beginning, the topic was discussed by many researchers around the world in terms of contemporary political and economic occurrences (Peternel and Grešš 2021; Triwahyuni 2022). For decades, heads of state and government, ambassadors, and diplomatic staff have been supporting businessmen and entrepreneurs in promoting trade and negotiating agreements.

Through an assessment of the literature related to the topic, it can be seen that there is no unanimous definition: "Economic diplomacy is mainly concerned with what governments do, in the broadest definition. It goes much wider than foreign ministries or any closed circle of officials. All government agencies that have economic responsibilities and operate internationally are engaging in economic diplomacy, though they might not describe it as such. (. . .) It is important to make clear at the outset that there is no single theory of economic diplomacy that can provide answers on how states, under given circumstances, will conduct policy" (Bayne and Woolcock 2011). Economic diplomacy is concerned with economic policy issues, e.g., the work of delegations at standard-setting organizations such as the World Trade Organization (WTO).

A country's economic diplomacy is hence based on its image since it is an effective way to create and facilitate the attraction of foreign investment and respective exports (António 2021). Economic diplomacy is fundamentally based on four vectors (Almeida 2015): 1. the use of political influence and strategic relations to promote international trade and investment, allowing the opening of new markets in diverse and distinct geographical areas; 2. improving the functioning of some specific market aspects and correcting existing failures, such as reducing the costs and risks of cross-border transactions (where property rights are included); 3. strengthening the mutual benefits of interstate cooperation and political-economic relations, so that it is possible to increase the security of trade and citizens; and 4. functioning as a long-term instrument for the resolution of bilateral and, crucially, multilateral conflicts.

Nowadays, economic diplomacy deals with issues of economic policy (macroeconomics), including bilateral relations between states and the work of economic offices of embassies in international economic and financial organizations (António 2021). There are three main categories of economic diplomacy: 1. regional economic diplomacy (Barston 2006), made between organizations of a regional nature: the European Union, the Organisation of African, Caribbean and Pacific States (OACPS), Mercosul, etc.; 2. bilateral economic diplomacy, carried out between two countries by heads of state and government, embassies and consulates, to boost their economic power at an international level the most traditional model is state-to-state relations; and 3. multilateral economic diplomacy, executed between representatives of various states in various international organizations such as the International Monetary Fund (IMF), the Organisation for Economic Cooperation and Development (OECD), the United Nations (UN), the World Bank, and the World Trade Organization (WTO), among others (Barston 2006).

Begeny (2018) noticed that internationalization is usually discussed in terms of being part of globalization. Hence, internationalization is defined as the response to globalization, which includes the sharing of knowledge, people, technologies, values, and ideas without borders. Globalization and internationalization are two different but closely interrelated concepts (Bužavaitė et al. 2019).

Currently, especially in countries with a small internal market, the internationalization process for SMEs is inevitable. With the expansion to new markets, decisions are more complex, and companies are naturally more vulnerable (the exposure carries more risks and more responsibilities). Internationalization is recognized as the main force for economic growth, competition, and new workplace creations (Bužavaitė et al. 2019). Singh et al. (2010) assumed that internationalization is provoked by globalization processes all around the globe and noticed that more and more small companies internationalize at earlier stages of their development. Thus, internationalization can be associated not only with exports but also with other ways of performing international trade or collaboration.

Theories on internationalization led to the need to identify the internal and external motives that stimulate the international growth of a company (Bolzani and Foo 2017). Internal motives can be related to a lack of resources or specialization in one product only. On the other hand, external motives could be associated with government support, promising orders from customers, and prevailing opportunities; thus, governmental support plays a significant role in the international expansion of SMEs (Shamsuddoha et al. 2009). The shortage of knowledge or resources in a company is the main obstacle to its internationalization, due to the uncertainty it may cause (Silva 2020).

The internationalization process develops in four sequential stages (Johanson and Vahlne 1977): (a) Stage 1: no regular export activities—the absence of regular export activities (indirect exports). Exports are occasional and allow companies to make first contact with the market without committing resources, but with the disadvantage that the information received is reduced; (b) Stage 2: exportation through agents—exports via agents who provide greater knowledge of the international market. However, these companies end up with a greater resource-related commitment (direct export); (c) Stage 3: establishment of an overseas sales subsidiary—establishment of local sales subsidiaries, which allow companies to directly control the information channel, with the disadvantage of increasing costs and risks; and (d) Stage 4: overseas production/manufacturing units—establishment of local production subsidiaries, which ultimately require a greater commitment of resources, compared with the other three stages (Johanson and Vahlne 1977).

Upon examination of these four stages, it is clear that the internationalization process matures as the level of experience and knowledge of the market and foreign operations increases, and, at the same time, an increasing allocation of the company resources in those markets is witnessed—the greater the information regarding the market that a company has, the more important the resources made available for market penetration (engagement with the market gradually grows stronger (Lin 2010)).

However, it is important to highlight the differences and commonalities regarding the internalization process in the GCC region and the Middle East and North African (MENA) countries. They are both composed of Arab countries; nonetheless, they behave differently.

Common Factors: (a) Both the GCC countries and other MENA countries have recognized the need to diversify their economies away from reliance on oil and gas revenues. This often drives internationalization efforts in sectors such as tourism, finance, and manufacturing. (b) Globalization has influenced both regions, leading to increased interconnectedness and trade opportunities. Both internationalization efforts are often driven by the desire to tap into global markets and attract foreign investment. (c) Investments in infrastructure, such as ports, airports, and transportation networks, are common across both the GCC and other MENA countries to facilitate trade and economic development. Varying Factors: (a) GCC countries typically have higher levels of resource dependency due to their significant oil and gas reserves. This can impact their internalization efforts, as they may focus on leveraging these resources for international expansion, while other MENA countries might prioritize different sectors due to their lack of abundant natural resources. (b) The size and maturity of domestic markets vary significantly between GCC countries and other MENA countries. GCC countries often have smaller populations but higher per capita income, whereas other MENA countries may have larger populations but lower levels of disposable income. This influences the scale and focus of internalization or internationalization efforts. (c) Geopolitical considerations play a significant role, particularly in the MENA region. GCC countries may have different geopolitical priorities and alliances compared to other MENA countries, impacting their internationalization strategies and target markets. (d) The regulatory environment, including trade policies, investment regulations, and legal frameworks, can vary between GCC countries and other MENA countries, affecting the ease of doing business and the attractiveness of each market for internalization or internationalization. (e) Cultural and social factors, such as language differences, consumer preferences, and business practices, can influence internationalization strategies differently in each region. GCC countries, for example, may share more cultural similarities due to geographic proximity and common language (Arabic), while other MENA countries may have more diverse cultural landscapes.

The GCC market encompasses about 58.6 million inhabitants (World Bank). However, it is notoriously difficult to gather statistics about the population of the ME and North Africa. Many countries' populations are very transient. The Gulf is changing exponentially in many ways. Some Gulf Arabs complain that it is changing too much, too fast. The Gulf countries are culturally influenced by the values, traditions, and practices of the Indian subcontinent as well as Iran. Several Islamic beliefs and practices will impact the rhythm of the business day, how companies conduct business, and how well they will be accepted by their local business partners, customers, and colleagues. To succeed in the ME, companies must do some research on Gulf history, culture, and religion (Williams 2017).

With the arrival of Western businesses that are new to the ME, some organizations have become very successful; others struggle, and some have failed. Often, the difference between a successful organization and one that fails is the organization's level of cultural intelligence. Cultural intelligence has never been more important as businesses globalize, especially in parts of the world that are very different from markets in the West. Cultural and social mistakes can cost businesses. Learning how to do business in the ME without offending is crucial.

The World Competitiveness Ranking, an annual report published by the International Institute for Management Development (version 2022), assesses the competitiveness of countries based on various economic and business indicators. It shows that the GCC countries are ranked among the top 40 countries: Bahrain is 39th, Qatar 18th, Saudi Arabia 24th, UAE 12th, Kuwait (no data available) and Oman (no data available).

It is important to bear in mind that each GCC country has unique characteristics, business practices, and regulations. Companies must tailor their approach to each specific market, build local relationships, and adapt to the cultural nuances to increase their chances

of success in the GCC region (Williams 2017). Also important to mention is that the viability of ports and airports plays a crucial role in attracting foreign business in the internationalization process. Access to efficient ports and airports facilitates the movement of goods, services, and people, which is, obviously, essential for international trade and investment. Ports serve as gateways for imports and exports, allowing for the seamless flow of goods between countries. Countries with well-developed ports can handle larger volumes of trade and accommodate larger vessels, reducing shipping costs and transit times. This makes them more attractive destinations for foreign businesses looking to expand their markets or establish supply chains. Similarly, airports are vital for international business travel and the transportation of high-value or time-sensitive goods. Countries with modern airports, such as the ones in GCC, offering extensive flight networks and efficient customs procedures, are better positioned to attract foreign investors, business travelers, and multinational corporations. Furthermore, the presence of solid service and financial sectors enhances the attractiveness of a country for foreign businesses. A robust service sector, including banking, insurance, legal, and consulting services, provides essential support to foreign companies operating in the country. Access to reliable financial services facilitates transactions, investments, and capital flows, contributing to a conducive business environment. In a few words, the viability of ports and airports, coupled with strong service and financial sectors, creates an environment that is highly conducive to attracting foreign business in the internationalization process of economic diplomacy. These factors enhance connectivity, facilitate trade and investment, and contribute to the overall competitiveness of a country on the global stage.

On the other hand, doing business in the GCC region is currently easier than in the MENA region. Proof of this is the fact that these countries are ranked higher in the World Competitiveness Report. Why?: (a) GCC countries, particularly those like the United Arab Emirates and Qatar, have diversified their economies beyond oil and gas. They have heavily invested in sectors such as tourism, finance, and technology. This diversification provides stability and resilience against fluctuations in oil prices, contributing to economic competitiveness. (b) GCC nations also have made significant investments in infrastructure development, including transportation, telecommunications, and energy—this infrastructure supports business activities, enhances connectivity, and facilitates trade, all of which contribute to competitiveness. (c) Those six countries often offer more business-friendly environments compared to other MENA nations—they have streamlined bureaucratic processes, implemented investor-friendly policies, and established free zones to attract foreign investment. These measures foster entrepreneurship and stimulate economic growth. (d) GCC countries have invested in education and training programs to develop their human capital. They attract skilled expatriates from around the world and provide opportunities for professional development. A highly skilled workforce enhances productivity and innovation, key components of competitiveness. (e) Compared to MENA countries, GCC nations generally exhibit greater political stability and security. This stability provides a conducive environment for businesses to operate and invest with confidence, contributing to overall competitiveness. (f) GCC countries benefit from their strategic geographic location, serving as important hubs for trade and commerce between East and West. Their proximity to major markets and access to global transportation networks enhance their competitiveness as regional economic centers. While GCC countries have all of these advantages contributing to their high rankings in competitiveness, it is important to note that challenges persist, including issues related to labor market dynamics, sustainability, and socio-economic inequality. Additionally, each GCC country may have a unique set of strengths and weaknesses that influence its competitive position.

International relations traditionally refer to the diplomatic interplay between sovereign states. Nevertheless, states are only one type of actor in international relations. International organizations, non-governmental organizations (NGOs), multinational enterprises (MNEs), and even terrorist groups are all also actors in international relations—these are the entities "acting" in international relations. International relations could not work without active

participation from its actors, where they all are obligated to interact with one another. Different types of international actors have different levels of sovereignty and different legal obligations; however, all actors in international relations are responsible to each other and international law. All of these actors interact with each other in an attempt to achieve their interests. Actors in international relations strive to realize their political goals and interests with other actors: "For a state, this might mean acquiring territory from another state through diplomatic or military means. For an intergovernmental organization, it might mean promoting the economic interests of a large group of otherwise sovereign states. For a multinational corporation, it might mean lobbying a state government to change labor or environmental laws". The peaceful management of international relations by state actors is called diplomacy (Reiss 2022).

## 3. Methodology

This study presents a comprehensive analysis of the internationalization process of economic diplomacy by exploring the experiences of renowned Portuguese companies within the Gulf Cooperation Council. The Gulf Cooperation Council countries are integral players in the Middle East market, characterized as rentier states that are heavily reliant on oil and gas revenues. They possess varying levels of economic and military strength, with Saudi Arabia and the UAE being the most prominent.

A focus group technique was applied to moderately sized groups based on individuals who have previously had some common experience or presumably share some common views, including a moderator (Yin 2011, 2012). Focus groups are one type of qualitative data collection, based on interviews, using a researcher-led group discussion to generate data (Given 2008). This methodology aims to obtain in-depth data from a purposely selected group of participants, with the number of participants per focus group (where reported) ranging from three to 21 participants (Nyumba et al. 2018).

To answer the questions with the qualitative methodology (Tong et al. 2007), this study was conducted with three Portuguese companies from different sectors, with a path in the ME market and currently with the highest turnover in the GCC market (2023):

(i) Company "A"—100% Portuguese company specializing in retail (brand design, exhibitions, among others). With 43 years of experience in business and investment acceleration, delivering fully integrated services, from strategy to execution, connecting the dots, etc., it started its internationalization process in 2012. Currently, it is present in six markets: Portugal, Spain, Brazil, Germany, China (where it has a strong presence), and, more recently (since April 2021), in the UAE. The professionals in this company go beyond today's challenges to anticipate the potential long- and short-term consequences of ever-changing business, financial, and technology strategies. They support their clients and partners in exploring potential obstacles to change and collaborate on critical decisions that can deliver real value to their businesses.

(ii) Company "B"—a family clothing and accessories company based in Portugal. With around 40 stores in the ME, it is one of the most important markets for the company. In the market for a long time, it is nowadays one of the biggest Portuguese investments in the ME. The company took its first steps through a dream shared by four brothers who, in November 1989, embarked on the incredible adventure of building a brand that was marked by a remarkable lifestyle. It started as a small menswear shop, situated in a busy street in the heart of Lisbon, and business prospered from then on. In 1997, they opened their first shop inside a shopping center (Colombo Shopping Centre in Lisbon, at the time the largest in Europe)—a great challenge that allowed the company to learn a lot from this experience (the first collection of shirts for the season was sold in only 15 days). It was in 2007 that the company started its international expansion into Spain and the United Arab Emirates, where the first shop was opened in the Dubai Festival City Mall, followed soon after (2008) by the opening of other shops in Oman and Bahrain. In 2010, they increased the number of shops in the ME to seven, and today they have more than 20 in the GCC countries. They are proud to be in 16 countries, with 115 shops. Men, women, and children

can experience the Sacoor Brothers world in Portugal, Romania, United Arab Emirates, Qatar, Bahrain, Kuwait, Saudi Arabia, Jordan, Palestine, Algeria, Indonesia, Malaysia, Singapore, Sri Lanka, Georgia, and South Africa. The brand is recognized in the major fashion capitals.

(iii) Company "C"—a multinational technology company headquartered in Lisbon, dedicated to improving the quality, convenience, efficiency, and security in government services, travel, border control, and all smart facilities through the development and implementation of integrated, user-centric, digital identity management solutions and services built upon trusted biometric tokens. It is a leading partner of the most prestigious airports, airlines, governments, and private entities with critical security and identification challenges, supporting them to optimize the identification and flow of travelers. In 2011, they attempted to enter the ME, specifically in the Saudi Arabian market—however, at that time, without success. They entered the ME market through Qatar at the end of 2012, where they installed e-gates at the "new" Doha International Airport (opened to the public in 2014, this airport was designed to host the 2022 Football World Cup). This company has two ways of entering ME markets: through international public tenders or association with a local partner—the Qatar market is used as a reference for expansion in the region. After the award of the tender in Doha, they experienced some difficulties, as they were not physically present in Qatar, and at the very beginning of 2015, they decided to open an office in Dubai as a hub for the ME. Today, they are market leaders in four GCC countries (Qatar, UAE, Bahrain, and KSA). They are also present at Baghdad Airport in Iraq.

The design of the focus groups considers the relationships of the participants with the theme under study. The following precautions were, therefore, taken: (a) access to the groups' constituent elements, (b) problem profile, (c) the perspectives and discursive capacities of the groups' constituents, (d) their identification with the problem under investigation, (e) ethics in data collection, processing, and use, (f) the possibility of recording the moment, (g) representation of the experience of the research process and the experiences of the constituent subjects of the groups, (h) the analysis process and, (i) production of a written report.

The instrument used was a focus group to explore a specific set of issues (Tong et al. 2007) related to economic diplomacy in the internationalization process. To this end, an interview script was constructed which allowed the moderator to steer the course of information collection. This technique allows researchers to listen to the views and socially constructed beliefs of the participants (Chen 2012), and they are relatively free to discuss the topics covered. The inclusion criteria were as follows: Portuguese companies with a presence in the ME, represented by (1) top management (C-level) and (2) middle management, in an age range of 20 to 60 years. Participants agreed to participate voluntarily after being approached by points of contact identified in each of the three companies who were responsible for recruiting participants who met the inclusion criteria.

After identifying the companies, there were six participants per company respecting the inclusion criteria listed above, totaling 18 participants. The profiles of the participants were characterized by considering three variables, namely age range, position, and education level, as follows:

P1    Executive director, market access and corporate affairs 41–50 Master's degree
P2    Executive business manager 41–50 Bachelor's degree (post-grad)
P3    Chief financial officer 51+ Bachelor's degree (post-grad)
P4    Global chairman 41–50 Secondary school
P5    Head of operations 41–50 Bachelor's degree
P6    Head of public relations 41–50 Bachelor's degree
P7    Finance manager 41–50 Bachelor's degree
P8    Project leader 20–30 Master's degree
P9    Brand manager 31–40 Bachelor's degree
P10   Chief of staff to the CEO & chairman 41–50 Master's degree
P11   Sales manager 31–40 Bachelor's degree

P12   Head of commercial department 41–50 Bachelor's degree
P13   Regional director–Middle East and Africa 51+ Master's degree
P14   Vice president–strategic sales and global partnerships 51+ Master's degree
P15   Head of global business development 41–50 Bachelor's degree
P16   Chief revenue officer 51+ Master's degree
P17   Project manager 41–50 Bachelor's degree
P18   Senior manager 41–50 Bachelor's degree

In the analysis of the academic degrees of the respondents, it is interesting that the sample goes against the study by Fundação Ricardo Espírito Santo (FFMS), which notes that only one-third of Portuguese companies have senior managers with higher education. The sample exceeds these figures; however, one person in the top management of one of the Portuguese companies with the most success in the ME has a global chairman with secondary education. It is also important to highlight that most of the top managers have only a bachelor's degree.

Participants who met the inclusion criteria were informed of the study objectives and the procedures that would be followed for data collection, namely through the Focus-GroupIt platform. The way the feedback session would take place was explained: seven questions were posed by the researcher, regarding which participants would be completely free to express their opinions. The focus group was conducted through the abovementioned platform, where all ethical standards were ensured, ensuring participant confidentiality regarding data collection. The focus groups were applied between March and May 2023. The material was transcribed into an Excel sheet with 11 lines and 20 columns. The material obtained was imported into the NVivo software program. For each participant, cases and attributes were created to characterize them and enable the individual consultation of their testimony.

The interest of companies in international trade is always motivated by a search for business success. For this very reason, we considered it pertinent to include the word success in the research questions formulated.

(1)   How do you understand the concept of economic diplomacy?
(2)   Why did you internationalize your brand/service?

Choose one or more options and justify:

i.     To diversify company markets.
ii.    Global brand exposure.
iii.   Access to a larger talent pool.
iv.    Potential for new revenue.
v.     A greater breadth of investment opportunities.
vi.    To outpace the competition.

(3)   Did you follow specific steps/strategy, or was it a random process?
(4)   Do you use the economic office of the Portuguese Embassies around the world (Portuguese Trade & Investment Agency–AICEP delegations) to help enter new markets? If yes, what kind of help did you use? Please specify.
(5)   Considering your company has more than one office around the world, for the financial growth of your company, is it part of your strategy to enter new markets or invest in the Portuguese market? In which market are you gaining the biggest profit?
(6)   Compared with other markets, what are the main challenges of working in the Middle East and why?
(7)   Do you consider the Middle East as a potential market for the future? Please justify bearing in mind the importance of the regional market, the climate for doing business, the population of young people, and the strategic location of the ME countries. Of the six GCC countries, which one would you highlight as the most prominent (Kingdom of Bahrein, State of Kuwait, Sultanate of Oman, State of Qatar, Kingdom of Saudi Arabia, or the United Arab Emirates)?

## 4. Results

### 4.1. Word Frequency Study

As an exploratory form of data, the 100 most frequent words in the document containing the transcription of all of the interventions (exact matches) with a minimum length of four letters were studied. The most significant words were subject to a text search (locating the references in which these words were used) to appreciate the context in which they were spoken.

The analysis procedures followed the stages of thematic analysis to identify, analyze, and report the themes that emerged from the transcripts of the participants' interventions (Braun and Clarke 2006, 2012). The categorization technique presupposes the transformation of the data to achieve a representation of their content. To this end, the following steps were performed (Resende 2016): (a) reading the texts obtained from all of the interventions, (b) initial categorization of the meaning units based on the script questions of all of the interventions, naming them according to the terms used in the research, their purposes, the experience and knowledge of the researcher, (c) reading by themes and categories, reviewing all categorized material, reconfiguring coding where necessary and identifying the emergence of new categories or their deletion, and (d) interpretation and writing up of the results for each theme and its substitutes. At this stage and after 'intimate familiarity' (Charmaz 2006) with the content of the focus group participants' narratives, an attempt was made to establish a sequential and coherent discourse of the issues addressed in the focus groups, taking care to highlight the opinions and considerations of the participants.

Analyzing the word cloud (Figure 1 and Table 1) and taking into consideration the words that refer to the present research, the importance of the three main ones becomes evident: (1) markets, (2) companies, and (3) countries. In the current research, these three main words attest to a high relevance recognized as the major outcomes, since they are the most consensual and similar in all questions for all focus group participants. The results of this focus group suggest (1) markets, (2) companies, and (3) countries as a cutting-edge dimension strictly associated with economic diplomacy in the internationalization process. From the point of view of the methodology and procedure of analysis, research was conducted on these words to inquire more deeply into the context in which they were expressed.

**Table 1.** Most frequently used words (above 50% of the weighted percentage).

| Word | Extension | Count | Weighted Percentage (%) |
|---|---|---|---|
| markets | 7 | 134 | 4.25 |
| company | 7 | 52 | 1.65 |
| countries | 9 | 47 | 1.49 |
| economic | 8 | 43 | 1.36 |
| region | 6 | 43 | 1.36 |
| business | 8 | 42 | 1.33 |
| Middle East | 10 | 37 | 1.17 |
| potential | 9 | 33 | 1.05 |
| revenue | 7 | 26 | 0.82 |
| investments | 11 | 25 | 0.79 |
| brand | 5 | 22 | 0.7 |
| diversify | 9 | 22 | 0.7 |

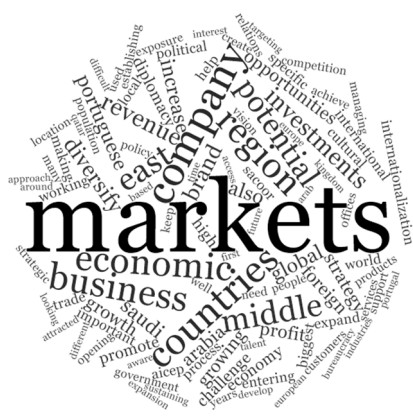

**Figure 1.** World cloud.

From the focus groups, one main theme, seven categories, and 26 subcategories emerged, representing the perceptions and considerations of economic diplomacy in the internationalization process (Figure 2).

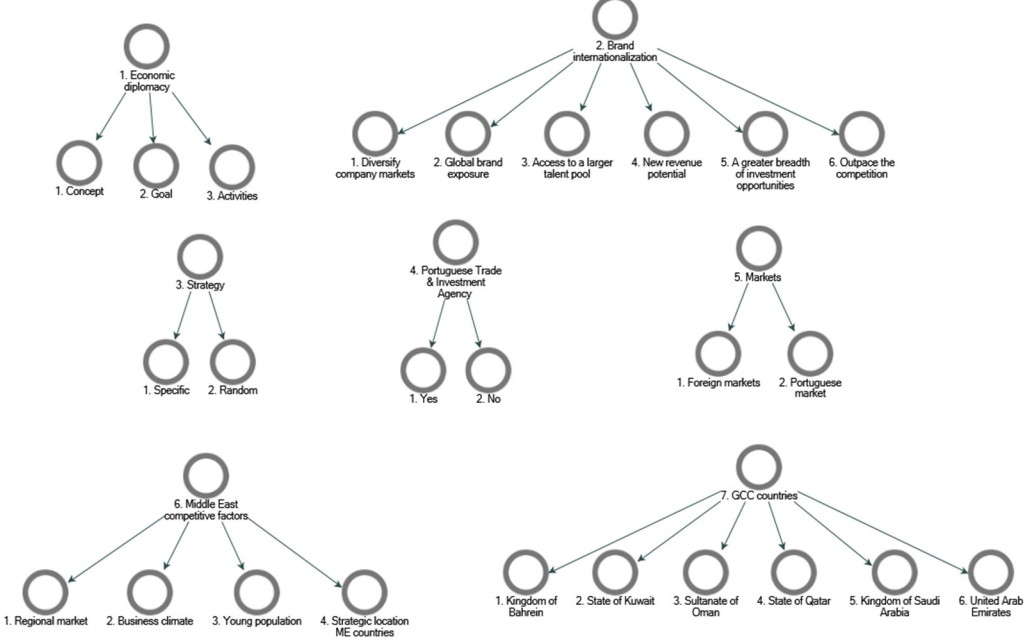

**Figure 2.** Themes, categories, and subcategories that emerged from the participant interviews.

Upon visualization of the data that emerged from the thematic analysis of the concept regarding the process of internationalization in the Middle East, seven themes emerged: (1) economic diplomacy, (2) brand internationalization, (3) strategy, (4) Portuguese Trade & Investment Agency, (5) markets, (6) Middle East competitive factors and (7) GCC countries. The "brand internationalization" factors presented a higher number of categories, totaling seven, followed by the "economic diplomacy" concept and "Middle East competitive factors", both with four sub-categories derived from one of the categories. The GCC countries were not considered as they are formally six countries—participants were expected to choose them.

Analysis of the units of meaning of each participant according to the category and subcategories allows a certain analytical flexibility and can enhance the transparency and trustworthiness of the qualitative research process. It can be inferred that the theme and underlying categories that achieved the most importance in terms of units of meaning for the participants were brand internationalization with 52, Middle East competitive factors

with 47, followed by the six GCC countries with 36, economic diplomacy with 33, new markets with 21, and strategy and Portuguese Trade & Investment Agency, both with 18.

### 4.2. Economic Diplomacy

Regarding the main purpose of this focus group—identifying the variables that make economic diplomacy effective in the Middle East and how it contributes to the internationalization of companies—it started with the definition of the concept of economic diplomacy that gathers four categories: concept, goals, activities, and economy.

### 4.3. Concepts

The participants defined economy diplomacy as 'government resources to promote the growth of a country's economy' (P1), defending the national interest (15) and using 'political institutions (diplomatic missions) in international economic relations' (P3), 'creating bonds among countries' (P8), the 'effort a country makes by using their diplomatic missions abroad' (P15) to 'help countries achieve the true essence of globalization' (P4).

### 4.4. Goals

- Promoting the economy (P6), developing investments and businesses (P9) and fostering growth (P10);
- Increasing exports, foreign investment, and tourism (P6);
- Facilitating better access to financing (P6);
- Creating bonds among countries (P8);
- Facilitating the entrance into a new market (P10);
- Allowing companies to gain some competitive advantage to economically penetrate foreign markets (P13);
- Defending the country's national interest (P15) economically speaking and promoting foreign policy objectives (P16);
- Boosting the country's competitiveness in the global economy and strengthening relationships with other countries (P17).

### 4.5. Activities

According to the participants, it is possible to identify which activities by which companies leverage existing diplomatic relations (P13) and are used when applying economic diplomacy. Economic diplomacy uses government resources (P1 and P11) (diplomatic channels, resources, and tools (P17)) plus power (P2) to promote the growth of a country's economic interests and achieve its economic objectives (P17). Activities such as trade negotiations, investment promotion, the use of sanctions, and other economic tools to advance diplomatic objectives (P3) are highlighted as facilitating the entrance (P10) of a company into a specific country where there may or may not be national diplomatic representation.

### 4.6. Brand Internationalization

The aim of one of the questions was to understand why a company internationalizes, taking into account six reasons (identified earlier, taking into account my experience, in loco, in the Middle East: (1) to diversify company markets; (2) global brand exposure; (3) access to a larger talent pool; (4) potential for new revenue; (5) a greater breadth of investment opportunities and (6) to outpace the competition. It is important to highlight that each respondent was free to choose as many reasons as they wanted.

With no doubt, "to diversify company markets" was the reason most chosen by the respondents, followed by " potential for new revenue" and "global brand exposure", with a significant difference compared with the other three indicators. The 18 respondents, as per the survey, did not consider the ME a place that gives access to a large talent pool.

Analyzing the open answers to this question, it is clear that the focus of internationalization of the three companies was to increase the company's revenue (P1, P3, P6), taking their services to other geographies, enlarging their commercial base (P13, P16, P17) and

reaching new customers to boost sales and increase profitability (P9). Bringing customers to Portugal (P4) was also highlighted by some participants. Diversifying company markets constitutes a risk-reduction strategy for the respondents to expand into new markets and achieve greater profitability (potential for new revenue).

With the current situation in Europe and the size of the Portuguese market (P14, P15), companies highlight the importance of seeking new markets as sources of revenue. The Middle East, a region with potential economic growth (P17), has huge potential (P5), seen as a huge shark tank (P2). Some of the respondents even said that internationalization is the only way to keep increasing revenue (P15, P18).

### *4.7. Strategy*

This question aimed to understand if companies followed a specific or random strategy when starting their internationalization process. As we would expect, most of the respondents told us they followed a specific strategy, considering the main characteristics of the ME (P4). Only a few stated that, in their opinion, entering the ME came about by chance due to unforeseeable events, such as the COVID-19 pandemic: "I during the pandemic years was an exception where (...) market access strategy is a bit compromised by the pandemic's impact on the market and local services" (P1), which led the company to enter the market earlier than planned (P5). Ultimately, what was the specific strategy used by companies? The respondents used the strategy defined in the business plans made depending on the market, its culture, and the social, political, and economic environment (P2), but with the resilience to quickly adjust and adapt (P4). It was made clear that all three companies started with a restrained strategy, and once they began to see results, they significantly increased their ambition (P8, P10).

According to the respondents' answers, they all consider the ME a particular market where the selection of local partnership (be it from technical vendors or pure agents) is essential to "open some doors" (P13)—a solid network, where a huge number of businesses are made through trustworthiness, is fundamental (P14). The wow factor is something that must frequently be in a company strategy when internationalizing to the ME (P17)—introducing modifications and adjustments to the products to be more customized to the local cultures is almost "mandatory" (P17).

### *4.8. Portuguese Trade & Investment Agency*

This, there is no doubt, as almost 100% of the respondents answered that they got in touch with AICEP during the internationalization process. The main reasons highlighted for companies getting in touch with AICEP are not only to introduce the company but also to create potential synergies, optimize their return on investment (ROI) in the region, and create value-based business relations (P1, P6). Companies request the AICEP to help with information on the business climate, bureaucracy, legal issues, sponsorship, building relationships with local people, consular affairs, and booking institutional meetings (P7, P12, P13, P14, P15).

The respondents also mentioned that what they expected from AICEP delegations around the world (they asked AICEP to increase their network (P18)) is different from one geography to another. According to their answers, in the ME, the three companies that took part in this study stated that they had direct support (with substantial success (P5)) in establishing contacts and providing sociocultural advice for the market (P4).

### *4.9. Market*

The answers were clear. Nowadays, the respondents have the feeling and the experience that, in European markets, it is difficult to make a profit, particularly in Portugal (a very small market with limited growth (P5, P15)) where the cost of living has increased significantly. As such, they all state that looking for foreign markets is very important to keep growing (P6, P13, P14, P18). However, some respondents say they have not stopped investing in Portugal (P6).

In the same question, it was possible to ascertain in which markets the three Portuguese companies with offices in the Middle East have the most profits: China, the Middle East (P12), particularly in Saudi Arabia, and Southeast Asia.

### 4.10. Middle East Competitive Factors

The ME is best known for oil production and export (it is home to 53% of the world's proven oil reserves and nearly half of all known natural gas reserves, with public policies implemented to diversify it (P6)), which significantly impacts the entire region through the wealth it generates. In recent years, many of the countries in the region have undertaken huge efforts to diversify their economies, creating opportunities for companies (P1) from several sectors of activity. Contrary to what is happening in Europe, birth rates remain high, and in the ME, a combination of a rapidly increasing youth population and an influx of migrant workers (searching for safety and business) has propelled countries forward, particularly the GCC countries. From the ME, Portuguese companies can target all Middle East and Southeast Asian markets. The UAE (especially Dubai and Abu Dhabi) continues to be a strategic business hub, with business-friendly free zones and a quickly growing economy that acts as a comfortable anchor where international companies settle to access Middle East and Southeast Asian markets (P1).

This question aimed to understand what the respondents have to say regarding four ME competitive advantages that they have experienced while living in the region:

1. **Possibility of negotiating in a regional market that is in the early stages of growth**

The ME is a large, diverse market with a population of over 400 million people. The region is also home to many fast-growing economies, such as the UAE, KSA, and Qatar (P2).

2. **ME business climate**

Many ME countries have been working to improve their business climates in recent years, making it easier for foreign companies to establish themselves and conduct business in the region (P2). Until June 2023, there was no corporate tax, and from July, a tax of only 9% will be applied in the UAE (P7).

Doing business in the ME revolves around personal relationships, family ties, trust, and honor (P6).

3. **Young population**

The ME has a large and growing young population, which represents a potential market for a wide range of products and services. This young population also presents an opportunity for companies to tap into a growing workforce and talent pool (P2). These countries have also many very experienced and skilled people from surrounding countries (especially India) working for a lower salary compared with Europe (P7, P10, P11, P17)

4. **Strategic locations for negotiation with the MEA markets**

The Middle East is strategically located between Europe, Asia, and Africa, connecting east and west, north and south, making it a hub for trade and commerce and providing a natural bridge between producers, manufacturers, and consumers of some of the world's most consumed commodities. The region's proximity to major oil-producing countries also makes it an important center for the energy industry (P2, P6, P13, P14). Important to mention is air connectivity: Emirates Airline, from the UAE, is connected to the world (P7, P8).

Also mentioned as competitive factors were investment opportunities, considering that the ME has been encouraging foreign investment in various sectors such as real estate, technology, renewable energy, infrastructure, and tourism, which provide promising opportunities for companies and investors (P2, P4, P18); its political stability; the "clear motivation to achieve the best that can be done in the world" (P4); the large-scale investment projects (P6, P9); and being a very safe place to live and work with good education and health services (P8, P10, P11).

However, respondents highlighted that the ME is a complex and dynamic region, with a wide range of political and economic conditions that can be different from country to country. Therefore, it is important for companies and investors to thoroughly research the specific market and country they are considering before entering the market. Companies must bear in mind that it is "impossible to do (good, sustainable) business in the region without fixing a team there" (P6).

Overall, the Middle East is a potential market for the future (respondents say "the Middle East is the future"), a very desirable location for business (P7) with a large and growing population, a strategic location, a diverse range of industries and a growing number of investment opportunities (P2, P3, P9).

It is important to mention that, upon analyzing all of the replies, this question was the one where the respondents gave the most complete answers. The enthusiasm all of the respondents showed about doing business in the ME was notable.

*4.11. GGC Countries*

Lastly, the respondents were asked about which of the six GCC countries they would highlight. The answers were very clear! Respondents consider Saudi Arabia as the "sexiest in the GCC" (P1), considered today as one of the biggest economic players in the world (P2). They intend to become even bigger with all of the new investments in real estate, tourism, technology, and international events (P2). It is the largest of the GCC countries, with 35.95 million people and currently strongly investing in diversifying the economy and luring new companies into its internal market (P6, P8, P17). The building of new cities from sand, artificial intelligence, ports, sustainable cities, and the "new marketing campaign: Cristiano Ronaldo" make this market very attractive (P7). Saudi Arabia is "the most prominent market that is worth the investment" (P15).

After Saudi Arabia, the UAE and Qatar are seen as having potential in the near term. Bahrein, Kuwait, and Oman were not mentioned by any interviewee.

**5. Discussion and Conclusions**

Discussion of the focus group results points to a set of main outcomes, boundaries, implications, advances, and directions. The main outcomes and closing remarks attest to the importance of economic diplomacy when entering new markets—in this specific case, the ME market. Undoubtedly, this study represents a considerable step forward in understanding not only the motivation for companies using the economic intelligence of the embassies (economic office) when entering new markets (taking the Portuguese case as an example) but also the increasing importance of the ME in helping companies to increase their turnover (Bayne and Woolcock 2017; Visser 2019).

All focus group participants advocated a clear understanding of the activities that governments need to undertake to achieve successful outcomes when "applying" economic diplomacy. It was clear that economic diplomacy involves the use of diplomatic tools and strategies to advance a country's economic interests in the international arena. To achieve successful outcomes, governments typically undertake various activities, including: (a) trade promotion and negotiation (governments engage in promoting exports, attracting foreign investment, and negotiating trade agreements to enhance market access for their goods and services), (b) investment facilitation (governments work to create a favorable investment climate by offering incentives, reducing barriers to investment, and providing support to both domestic and foreign investors); (c) diplomatic engagement (diplomatic channels are used to foster relationships with other countries, address trade disputes, and advocate for the interests of domestic businesses abroad); (d) policy coordination (coordination between various government agencies, including trade ministries, foreign affairs departments, and economic development agencies, is crucial to ensure a coherent and effective economic diplomacy strategy): (e) capacity-building (providing training and support to domestic businesses, particularly SMEs, helps them navigate international markets and capitalize on economic opportunities); (f) public diplomacy (governments use public

diplomacy initiatives, such as cultural events, media outreach, and educational programs, to shape perceptions abroad and build support for their economic agenda); (g) international cooperation (collaborating with other countries and international organizations on issues such as trade facilitation, standardization, and infrastructure development can create mutually beneficial outcomes and strengthen diplomatic ties). Furthermore, based on this assumption, to increase their revenues, companies choose to diversify their markets, i.e., internationalize their brand, by using a specific strategy, especially in the ME, where the selection of a local partnership is essential to "open some doors". Considering this, almost 100% of the participants got in touch with the AICEP ME during the internationalization process to understand better the market and to create potential synergies and value-based business relations. ME competitive factors are extremely clear for the participants, not only the four factors identified earlier in the research, but it is also important to bear in mind the various investments that are being made by GCC states in their own countries. Participants look to the ME as a potential market for the future, a desirable strategic location for business with a growing number of investment opportunities. Saudi Arabia is, according to the focus group participants, undoubtedly the most promising market for the future (AICEP 2023; Fernandes and Forte 2022; Lederman et al. 2010; Pacheco and Matos 2022; Rede Diplomática 2023).

A stream of various research analyzed in the dominant literature shows that the concept of economic diplomacy has been changing over time, along with the concept of globalization. The meaning of these concepts seems quite clear for companies that understand it as "the effort a country makes by using their diplomatic missions to help countries to achieve the true essence of globalization"—an essential tool to help companies enter new markets. It has been proven that this concept remains closely associated with government intervention to promote companies outside their borders. Sometimes, the first connection that companies have with the steps of internationalization is through their trade and investment agencies. It was evidenced that although companies sometimes enter a new market by chance, most of the time, a strategy can be well designed by having a meeting with the embassy's economic office. Nowadays, the information available when entering a new market is so broad that sometimes companies might get lost, and here is the point where trade and investment agencies can be useful to guide first steps. Moreover, it can also be inferred that for Portuguese companies, entering new markets is a must to make a profit and keep growing (Adkhamjonovich 2022; Bayne and Woolcock 2017; Moons and van Bergeijk 2016; Yakop and van Bergeijk 2011).

Based on the aforementioned results for the GGC countries, diplomacy has become an essential instrument for countries and companies in the context of international trade, globalization, and competition; it is a key aspect of a country's external relations and has been practiced for decades, involving the support of diplomats and government officials in promoting trade and negotiating agreements, as also noted by renowned authors (e.g., Yakop and van Bergeijk 2011; Peternel and Grešš 2021). Moreover, economic diplomacy focuses on supporting companies, attracting foreign investment, and transforming embassies into hubs for economic activities abroad. Economic diplomacy relies on a country's image and its ability to attract foreign investment and promote exports, making it an effective tool for international economic engagement, corroborating. Therefore, the findings of this study are in line with previous recognized studies (Bordón and Alrefai 2023) in another similar perspective, namely Saudi Arabia's economic diplomacy through foreign aid relating dynamics, objectives, and mode.

One of the major contributions of this study is the understanding of the importance of economic diplomacy when a company wants to internationalize to the ME. It allows them to create potential synergies, optimize their ROI in the regions, and create value-based business relations. They have a range of free services from the government that can be seen as the first steps in internationalizing, having the seal of trust based on the experience that each embassy collects over the years. Another main conclusion of this study is the clear evidence of understanding the purposes, the challenges, and the potential of the

ME market for companies. Without a doubt, it was clear that companies internationalize to "diversify company markets" and look for "potential for new revenue", taking into account that the ME is a significant wealthy geographic area with countless opportunities waiting to be grasped. But, as with all regions, the ME has a panoply of challenges that make this market less easy to enter if a company does not study it carefully (here is where economic diplomacy can play an important role). To sum up, economic diplomacy allows international firms to create synergies and optimize their ROI and business relationships.

Regarding the implications for companies in economic diplomacy, stakeholders of trade and investment agencies should be aware that the head of the economic office changes on a three- to four-year basis, and sometimes, when a new head arrives, he needs time to reactivate the contacts and understand the market. Regarding limitations, several general research limitations can be considered, namely: (a) complexity and diversity of the Middle East; (b) political sensitivity and access; and (c) time and resource constraints.

To address the research limitations and enhance the comprehensiveness of our analysis, we acknowledge the necessity of expanding our scope to encompass the multifaceted dimensions of the MENA region. This includes delving into the economic intricacies stemming from varying levels of development and resource endowments, as well as exploring the demographic diversity shaped by factors such as youth bulges, migration patterns, and urbanization trends. Moreover, recognizing the rich cultural tapestry and the political complexities inherent in the region will further enrich our understanding, enabling us to provide more nuanced insights for both seasoned scholars and readers less familiar with the MENA context.

Future research proposals for economic diplomacy can explore various dimensions and areas of interest. Toward this end, the development of a model for a company to enter each GCC market is suggested, seen as a guide for companies to take their first steps in the Middle East. All of these research proposals provide a starting point for further exploration of economic diplomacy. Researchers can select specific areas of interest and tailor their studies to address specific regions, industries, or policy contexts.

**Author Contributions:** Conceptualization, and methodology, D.P.; software, validation, and formal analysis, V.S.; investigation, resources, data curation, O.S.; writing—original draft preparation, writing—review and editing, S.W.; visualization, supervision, project administration, and funding acquisition, R.F. All authors have read and agreed to the published version of the manuscript.

**Funding:** This research received no external funding.

**Informed Consent Statement:** Not applicable.

**Data Availability Statement:** Data Availability Statements are available with authors.

**Conflicts of Interest:** The authors declare no conflicts of interest.

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
