# Peer review of "Towards Internationalization: Exploring Economic Diplomacy in the Middle East (GCC)"

_economies, doi:10.3390/economies12040082_

Round 1

Reviewer 1 Report (Previous Reviewer 1)

Comments and Suggestions for Authors

The Internationalization Process of Economic Diplomacy: A Case Study of the Middle East

Economies

(Manuscript ID: economies-2924004)

The authors of this manuscript entitled “The Internationalization Process of Economic Diplomacy: A Case Study of the Middle East” (Manuscript ID: economies-2924004) have addressed my previous comments and suggestions. The paper requires careful English editing before publication. 

Comments on the Quality of English Language

The paper requires careful English editing before publication.

Author Response

Thank you. We made the proofreading process.

Reviewer 2 Report (Previous Reviewer 2)

Comments and Suggestions for Authors

Second review report for the manuscript:

The internationalization process of economic diplomacy: a case of the Middle East

The paper has been improved. However, the main Authors’ argument in their response to my review is that the previously published papers were organized in the same manner (?) Indeed, I prefer more comprehensive quantitative analyses and maybe I underrated this qualitative research.

·         I support my comment that the title is too general and it only partially reflects what is the paper about. The focus group is rather narrow since it comprises only three Portuguese companies and six participants per company. However, the conclusions are very general.

·         Moreover, I support the remark that the results from the word cloud are trivial and words reported in Table 1 (not in Table 2, because there is only one table in this paper – a typo in line 497) are rather obvious (markets, company, countries). The program that generates word clouds is usually used by students.

Author Response

Thank you for your comment and we respect your opinion. The previously published articles are in fact organized in the same way. Thank you.

Thank you for your feedback. Regarding the title, we developed another one: “Towards Internacionalization: Exploring Economic Diplomacy in the Middle East (GCC)”.

As for the focus group, in our scientific context, field of action and research, it is recommended to have an average of six participants in a focus group (to ensure greater robustness and representative to guarantee and ensure greater robustness and representativeness and is within the parameters advocated by the reference authors in focus groups as justified in the article ess). The conclusions have been adjusted.

Thank you for your feedback. We understand your point of view. Please note that word clouds also are usually is also in renowned papers by senior researchers.

Reviewer 3 Report (Previous Reviewer 3)

Comments and Suggestions for Authors

The authors have amended many of the suggestions made for their first submission.  However, I see that there are still neglected aspects in their manuscript that should be revised. 

Especially in the bibliography, for example, references 14 and 15 are the same.

14. Bordón, J., & Alrefai, E. (2023). Saudi Arabia’s foreign aid: the singularity of Yemen as a case study. Third World Quarterly, 0:0, pp. 1–18. 849 .

The first reference cited in the manuscript, (Garlick & Havlová 2020), I cannot find in the references.

Nor do I find (Choi 2023) in the references.

Nor are all these from the introduction in the references section:

(Karavar 2021)

(Triwahyuni 2022)

(Zelicovich 2023)

(Croatia et al. 2019)

(Khmel & Tykhonenko 2020)

(Wang 2020)

My opinion is that the literature cited is not connected to their study.

Unfortunately after not finding all these references cited, I do not further evaluate the manuscript. I believe that the manuscript should be rejected as it does not meet the minimum standard of a scientific publication, which is to cite authors correctly and conveniently. 

Comments on the Quality of English Language

I take no position on this aspect

Author Response

Reviewer 3 Comments

Changes

The authors have amended many of the suggestions made for their first submission.  However, I see that there are still neglected aspects in their manuscript that should be revised.

Sorry for that.

Especially in the bibliography, for example, references 14 and 15 are the same.

14. Bordón, J., & Alrefai, E. (2023). Saudi Arabia’s foreign aid: the singularity of Yemen as a case study. Third World Quarterly, 0:0, pp. 1–18. 849 .

We corrected the references. Thanks.

The first reference cited in the manuscript, (Garlick & Havlová 2020), I cannot find in the references.

We corrected the references. Thanks.

Nor do I find (Choi 2023) in the references.

We corrected the references. Thanks.

Nor are all these from the introduction in the references section:

(Karavar 2021)

(Triwahyuni 2022)

(Zelicovich 2023)

(Croatia et al. 2019)

(Khmel & Tykhonenko 2020)

(Wang, 2022)

We corrected the references. Thanks.

My opinion is that the literature cited is not connected to their study.

We understand you point of view. However, after the changes provided, we understand that the literature cited are connected to the study, due to the fact correlating the main topics of economic and business diplomacy as well as internationalization. Thanks.

Unfortunately after not finding all these references cited, I do not further evaluate the manuscript. I believe that the manuscript should be rejected as it does not meet the minimum standard of a scientific publication, which is to cite authors correctly and conveniently.

We appreciate your comments and your decision. However, we understand that it is important to have the opportunity to correct mistakes. Once again, thank you for your valuable comments.

This manuscript is a resubmission of an earlier submission. The following is a list of the peer review reports and author responses from that submission.

Round 1

Reviewer 1 Report

Comments and Suggestions for Authors

The Internationalization Process of Economic Diplomacy: A Case Study of the Middle East

Economies

(Manuscript ID: economies-2885464)

This is an interesting paper that examines the internationalization process of economic diplomacy by looking at the expansion of Portuguese companies in the Gulf Cooperation Council (GCC) region. This study applies qualitative analysis, and the results highlight the significance of economic diplomacy in the internationalization process of foreign companies in the Middle East and North African (MENA) region in general and in the GCC region in particular. This well-structured paper covers a critical subject relevant to regions aiming to diversify their economies. I have a few comments and suggestions.

·       The author(s) could provide additional notes on the distinction between the internationalization process in the GCC region and the internationalization process in other MENA countries, noting a few common and varying factors.

·       The fact that the GCC countries are ranked high in the World Competitiveness Report, compared to other MENA countries, warrants a brief discussion.

·       I assume that there are factors that go beyond the abundance of oil and gas in GCC countries. Other MENA countries (such as Algeria and Iraq) are rich in natural resources but are ranked relatively low on the World Competitiveness Report. 

·       It would be interesting to briefly note how the viability of ports and airports and the presence of solid service and financial sectors contribute to attracting foreign business in the internationalization process of economic diplomacy.

·       One important concluding point is that economic diplomacy allows international firms to create synergies and optimize their ROI and business relationships. I suggest emphasizing further those important points.

·       The concluding section mentions that there are research limitations and that the research analysis should cover the economic, demographic, and cultural diversity and political complexity in the MENA region. I suggest adding a couple of sentences that clarify these points for unfamiliar readers. 

Comments on the Quality of English Language

 I suggest minor editing of the English language. 

Reviewer 2 Report

Comments and Suggestions for Authors

Review report for the manuscript:

The internationalization process of economic diplomacy: a case of the Middle East

The main contribution of the paper is not clear. Moreover, the research is only empirically-oriented and it does not contain any theoretical or methodological contribution. Only existing and simple methods are used.

Some additional major and minor comments:

·         The title of the manuscript is not proper and rather confusing. The Abstract explains that the paper investigates Portuguese companies (?) The title is decidedly too general and it does not reflect what is the paper about.

·         As mentioned above, the contribution of the study is not clear and it seems that this contribution is only an empirical one. Moreover, this is not clear what is really new in this research. The contribution should be better explained in the Introduction.

·         The results from the word cloud (presented in Figure 1 and Table 1) are trivial, and words reported in Table 1 are rather obvious.

·         In general, the presented qualitative and quantitative analyses are trivial, and these analyses are suitable for a student’s work rather than for scientific research.

Reviewer 3 Report

Comments and Suggestions for Authors

The manuscript titled "The internationalization process of economic diplomacy: a case study of the Middle East" addresses an important topic regarding the internationalization efforts of Portuguese companies within the Gulf Cooperation Council (GCC) countries, it exhibits several significant deficiencies:

The abstract briefly mentions the economic characteristics of the GCC countries as rentier states heavily reliant on oil and gas revenues, but it fails to provide a comprehensive overview of the socio-political dynamics and challenges within the region that may impact the internationalization process.

It lacks clarity regarding the specific objectives and methodologies employed in the study. It briefly mentions the internationalization process, economic diplomacy, and the Middle East market without clearly articulating the research questions, hypotheses, or theoretical framework.

The manuscript does not specify how the findings of the study advance existing knowledge in these areas. Without a clear articulation of the study's contribution, it is challenging to assess its significance.

The manuscript it does not specify the nature of these recommendations or how they can be practically implemented.

While the conclusions touch upon various aspects of economic diplomacy and internationalization in the Middle East, they lack clarity and precision in conveying specific findings and insights derived from the study. The conclusions would benefit from a more structured and focused presentation of key takeaways.

The conclusions do not reflect on the strengths and limitations of the methodology employed in the study. A critical reflection on the reliability and validity of the findings, considering the use of focus groups and qualitative research methods, would enhance the credibility of the conclusions.

Comments on the Quality of English Language

The text employs a diverse and specialized vocabulary. The sentence structure and grammar are generally solid, although there may be some areas where clarity and coherence could be improved.
